# Evaluating the association between COVID-19 and psychiatric presentations, suicidal ideation in an emergency department

**Michal J. McDowell**[1,2]*, **Carrie E. Fry**[3], **Mladen Nisavic**[1], **Mila Grossman**[1,2], **Charles Masaki**[1,2], **Emily Sorg**[1], **Suzanne Bird**[1], **Felicia Smith**[1], **Scott R. Beach**[1]

1 Department of Psychiatry, Massachusetts General Hospital, Boston, MA, United States of America, 2 McLean Hospital, Belmont, MA, United States of America, 3 Department of Health Policy, Vanderbilt University School of Medicine, Nashville, TN, United States of America

* mjmcdowell@mgh.harvard.edu

## Abstract

### Objective

To estimate the association between COVID-19 and Emergency Department (ED) psychiatric presentations, including suicidal ideation.

### Methods

Using an interrupted time series design, we analyzed psychiatric presentations using electronic health record data in an academic medical center ED between 2018 and 2020. We used regression models to assess the association between the onset of the COVID-19 outbreak and certain psychiatric presentations. The period February 26–March 6, 2020 was used to define patterns in psychiatric presentations before and after the coronavirus outbreak.

### Results

We found a 36.2% decrease (unadjusted) in ED psychiatric consults following the coronavirus outbreak, as compared to the previous year. After accounting for underlying trends, our results estimate significant differential change associated with suicidal ideation and substance use disorder (SUD) presentations following the outbreak. Specifically, we noted a significant differential increase in presentations with suicidal ideation six weeks after the outbreak (36.4 percentage points change; 95% CI: 5.3, 67.6). For presentations with SUD, we found a differential increase in the COVID-19 time series relative to the comparison time series at all post-outbreak time points and this differential increase was significant three weeks (32.8 percentage points; 95% CI: 4.0, 61.6) following the outbreak. Our results estimate no differential changes significant at the *P* value < 0.05 level associated with affective disorder or psychotic disorder presentations in the COVID-19 time series relative to the comparator time series.

### Conclusions

The COVID-19 outbreak in Boston was associated with significant differential increases in ED presentations with suicidal ideation and SUD.

**Data Availability Statement:** There are ethical restrictions on sharing our data set, as our data contain potentially identifying and sensitive patient

information per the Partners Institutional Review Board. Data is available upon request. Requests may be sent to the Partners Institutional Review Board protocol administrator, Virginia Rodriguez, at VGRODRIGUEZ@PARTNERS.ORG. For reference, our protocol number for the review board is 2020P—1482.

**Funding:** Author: MM Grant #: R25 MH094612, Funder: National Institute of Mental Health Website: https://www.nimh.nih.gov/index.shtml Sponsor did not play any role in the study design, data collection and analysis, decision to publish, or preparation of the manuscript.

**Competing interests:** No authors have competing interests.

## Introduction

The coronavirus disease 2019 (COVID-19) has emerged as a significant driver of physical morbidity and mortality worldwide, and it stands to exact sizable psychological toll. Psychological distress surrounding COVID-19 related illness and death may be exacerbated by fear, isolation, and physical distancing [1, 2]. Further compounding these factors are job loss and financial stress, which are well-recognized risk factors for suicide [1, 3]. Past global health crises offer cautionary lessons about potential mental health needs in the setting of a pandemic. During the 2003 severe acute respiratory syndrome (SARS) outbreak, stress was significantly higher in SARS-infected patients than in healthy controls [4]. Previous pandemics were also associated with escalations in suicidal ideation, suicide attempts, and completed suicides: rates of death by suicide in the United States increased following the 1918 influenza pandemic [5], and there was a 30% increase in suicide rates in older adults in Hong Kong during the 2003 SARS epidemic [6].

Particular to suicidal ideation, the coronavirus pandemic carries with it known risk factors for suicide at each level of the socio-ecological model [7–9]. The pandemic led to economic downturn, barriers to health care access, deprioritization of mental health in the setting of infection control measures, decreased access to community and religious supports, interpersonal conflict, unemployment, poverty, loneliness, and hopelessness [10–12].

Thus far, data on the association between COVID-19 and psychiatric illness includes case reports and case series, as well as data from anonymized health records, surveys, and publicly available entities [13–17]. Among psychiatric illnesses described as occurring or worsening in the setting of the pandemic, depression, anxiety, post-traumatic stress disorder, substance use disorders (SUDs), and general distress are all represented [3, 18–20]. One study found a twofold increase in emergency department visits for opioid overdoses as compared to the previous year during the first four months of the COVID-19 pandemic [21].

As one of the first states to suffer a major coronavirus outbreak in the United States, Massachusetts serves as an important test case in assessment of the relationship between the global pandemic and psychological distress. On February 1, 2020, Boston reported its first confirmed case of COVID-19, and by March 10, Governor Charles Baker declared a state of emergency in response to the coronavirus spread [22]. We used the period following the initial outbreak of COVID-19 in Massachusetts to assess the association between the pandemic and patients presenting with certain psychiatric complaints (a number of selected mood, psychotic, and substance use disorders, as well as suicidal ideation) to a tertiary care hospital Emergency Department (ED).

## Methods

### Study design

In this study, we employed a two-group, two-period cross-sectional (i.e., between-subject) study design, using data from electronic health records (EHR) of patients seen in the ED at Massachusetts General Hospital, a large, tertiary care hospital in Boston, Massachusetts, United States. The Partners Institutional Review Board approved this work (approval #2020P001482) and waived the requirement for informed consent. We used an interrupted time series design to compare changes in the proportion of patients seen with certain psychiatric presentations before and after the COVID-19 outbreak. We aimed to quantify the relationship between COVID-19 and the associated economic shut-down and quarantine orders on psychiatric ED presentations.

Comparative interrupted time series (CITS) design uses data collected over time, usually recorded at regular intervals [23]. We used a two-group (pre- and post-exposure), two time

period (2018–2019 and 2019–2020) CITS [24] study design for a number of reasons. First, adequately capturing the evolution of certain psychiatric presentations before COVID-19 reduces the chance of history bias [25]. Second, a comparison series allows for adjustment for trends unrelated to the coronavirus pandemic. For example, presentation of some psychiatric disorders varies throughout the year [12, 26]. To capture changes in psychiatric presentation associated with COVID-19, we used changes in presentation in the previous year as our comparator. These data can identify underlying trends, and, when an exposure occurs at a known time, post-exposure trends can be studied to assess changes from pre-exposure trends. The change in the comparisons series thus can serve as the counterfactual, or what would happen regardless of the exposure. Third, unlike a standard difference-in-differences approach, general CITS design allows for modeling a pre-period trend differential across the two groups. Finally, we used a general CITS, rather than a linear CITS, to minimize the parametric assumptions made regarding the outcome's evolution. A linear CITS assumes that the outcome is evolving linearly in the pre- and post-period [24]. On the other hand, general CITS assumes that any difference between the two groups is evolving linearly but does not make any assumption about the outcome's parametric form (i.e., does not assume a linear, quadratic, or other form). Additionally, we conducted an event study to visually assess the plausibility of these assumptions and found differential trends (i.e., non-parallel) in the pre-period suggesting that a standard difference-in-difference approach would not be appropriate.

We collected cross-sectional data from the hospital EHR for patients who received care from the ED psychiatry consult service. In this ED, patients receive emergency psychiatric consultation at the request of Emergency Medicine providers, and generally in response to presentations related to acute risk to self or others, significantly altered mental status and/or behavior, and/or presentations with extreme psychological distress. Data for the comparison time series were recorded from charts of patients seen December 2018 through May 2019. Data for the time series assessing the coronavirus outbreak were recorded from charts of patients seen December 2019 through May 2020.

To reduce chart burden, we abstracted charts on Mondays, Wednesdays, and Fridays of each week in the two study periods. Three days per week provided adequate power to detect an effect. Our data are at the person-day level. Patients were included if they received a psychiatric consultation in any one of the four observation periods (pre- and post-period for comparator time series and pre- and post-period for COVID-19 time series). Patients were allowed to appear in more than one of these four study periods. We adjusted the models for the total number of ED visits per patient per study period. Patients that did not receive a psychiatric consultation (even if one was requested) were excluded from this study. Additionally, patients with incomplete demographic or diagnostic records were excluded from analyses that used the characteristic of interest, including CITS regression modeling (approximately 90 records).

Race, ethnicity, gender, insurance, and legal status were available directly from statistical information maintained by the ED triage team and recorded in the EHR. Of note, ED staff received education in the July 2019 department newsletter regarding the importance of capturing race and ethnicity data accurately. Additionally, reminders were sent about the importance of accurately capturing these data during the initial COVID-19 wave in Boston. Psychiatric presentation diagnosis and presence of suicidal ideation were abstracted by clinician author chart review.

We defined the exposure as the time period from February 26, 2020 –March 6, 2020 to indicate the coronavirus outbreak in Massachusetts and the initial public health response. We refer to this period in this study as the "exposure." This period accounts for the growth in cases across Massachusetts, potentially associated with the Biogen conference held in Boston, which began on February 26, 2020 [27]. The first patient with a documented case of COVID-19 in

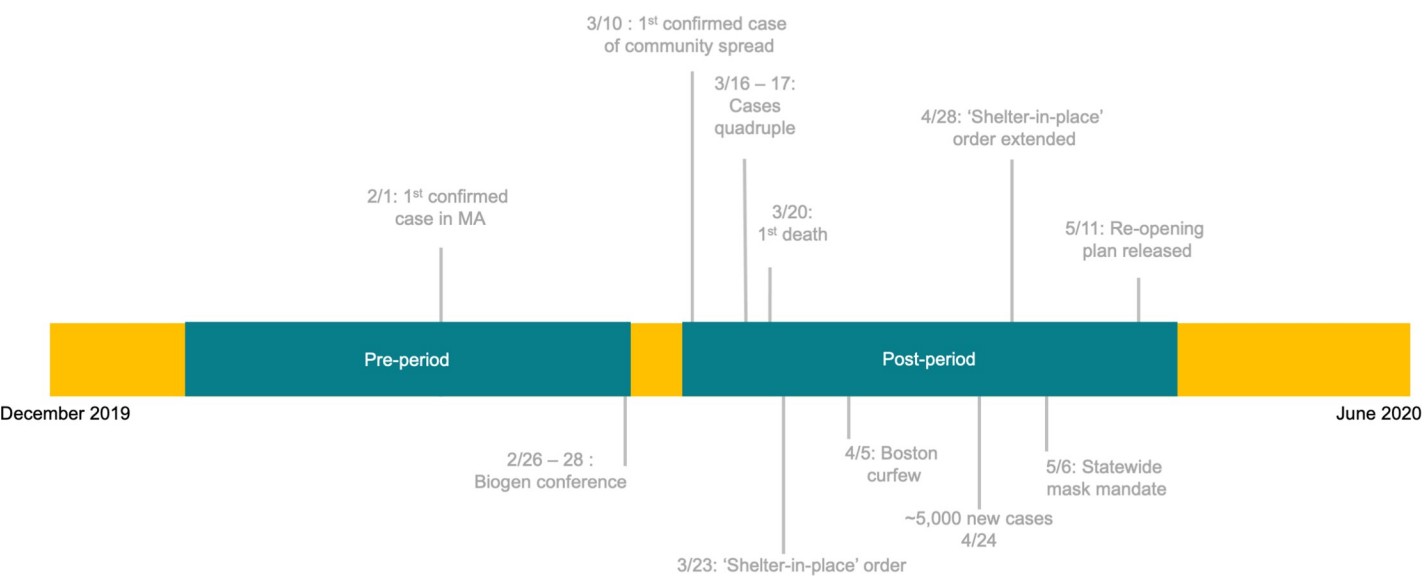

**Fig 1. Study period timeline.**

Massachusetts presented to the study site hospital on March 1, 2020. The comparator series pre-period runs December 24, 2018–February 25, 2019 and the COVID-19 series pre-period runs December 23, 2019–February 24, 2020. The post-period for the comparator time series is March 11, 2019–May 13, 2019 and for the COVID-19 series is March 9, 2020–May 17, 2020. The timeline of COVID-19 cases in Massachusetts and actions taken to mitigate spread of the virus, as well as the study period (including the exposure), are detailed in Fig 1.

## Outcome measures

From each medical record, we extracted information both automatically on patient demographics, as well as manually regarding psychiatric consultation findings. In other words, the medical records could be queried for summary information for some variables, but information had to be manually extracted by searching each record individually for other variables. Charts were reviewed by six psychiatrists (three residents and three attendings), and chart reviews that were ambiguous were adjudicated by other members of authorship team at bimonthly team meetings. The presence or absence of suicidal ideation as determined by text in the psychiatric consultation note, as well as up to three psychiatric diagnoses (not including SUDs), were recorded for each patient presentation. Reviewers included relevant and current diagnoses identified in the consult notes, as well as interpreted symptoms mentioned with Diagnostic and Statistical Manual of Mental Disorders 5th edition (DSM-5) criteria in mind.

Our outcomes of interest were the proportion of patients who presented with suicidal ideation (SI), SUDs (including alcohol, opioid, cocaine, and stimulant use disorders, as well as substance-induced mood and psychotic disorders), psychotic disorders (including schizophrenia-spectrum illnesses and brief psychotic disorder), and affective disorders (including major depressive disorder, and bipolar affective disorder) among all patients seen by the psychiatric consult service. We only included present suicidal ideation in our suicidal ideation outcome variable. For psychiatric diagnoses, we generally used the diagnosis given by the evaluating clinician. Historical labels were only used if they were deemed correct by the evaluating clinician researcher and relevant to the current presentation. For SUDs, historical diagnoses were used if substances were involved in the current presentation or if there was compelling evidence of

a current SUD in the consult note. We included suicidal ideation, SUDs, psychotic disorders, and affective disorders as our outcomes of interest because, based on clinical experience, these diagnoses are reliably and consistently captured in this ED setting. Anxiety disorders, although coded by the reviewers, represent a small proportion of ED visits and unless they are the primary reason for presentation, they are less likely to be consistently captured.

### Data analysis

First, we compared demographic and clinical characteristics among the pre-period of each time series (December 24, 2018–February 25, 2019 and December 23, 2019–February 24, 2020) using a two-sided t-test to measure baseline similarity between the two groups. Then, we estimated linear probability models (LPMs) (*f* lines) for each of our four outcomes. LPMs are preferred to logit or probit regression models with binary outcomes for both CITS and difference-in-difference models, due to the ease of interpreting interactions in LPMs relative to logit or probit models [28]. Additionally, LPMs estimate unbiased and consistent treatment effects for events that occur with a probability between 0.20 and 0.80. Regression models were adjusted with patient age, sex, insurance, and total number of psychiatric consults. Additionally, we adjusted with the patient's race and ethnicity because of the disproportionate impact of COVID-19 on Black and Latinx communities as a result of structural racism [29, 30]. We also included day-of-week and month fixed effects to account for the evolution of psychiatric presentations over the course of a week and year. We do not present data not adjusted by demographic factors, as the demographic composition of cohorts may change in different ways in the pre- to post-periods.

For suicidal ideation, we also adjusted for the presence of an affective disorder, psychotic disorder, and SUD. As a sensitivity analysis, we estimated the same regression for suicidal ideation in S3 Table, omitting the other diagnostic categories as covariates and adding back one of the diagnostic categories, since they may be on the causal pathway between the coronavirus exposure and suicidal ideation.

We estimated a treatment effect for each week in the post-period, rather than an average, to account for the temporal changes in care-seeking behavior during and after the initial surge in COVID-19 cases in Boston. In other words, our regression specification included an indicator variable for each post-period week. We interacted the indicator variable with the treatment variable, and each of these interactions represents the differential change for the week between the COVID-19 series and comparator series, as compared to the average pre-period levels. Additionally, we checked for autocorrelation in the time series and found no evidence of suspected autocorrelation structures, including AR1 or AR3. Additionally, the statistical significance of our results did not change using the most conservative method of multiple comparison adjustment, Bonferroni corrections. Our analyses used a 2-sided $P<0.05$ value as a threshold for reporting statistical significance and were performed using R statistical software.

## Results

To assess overall volume of ED visits requiring psychiatric consultations, we compared unadjusted trends in consults from the comparator (March-May 2019) post-period with consults from the COVID-19 (March-May 2020) post-period. This comparison showed a 36.2% decrease in psychiatric consultations (Fig 2) (561 ED visits requiring psychiatric consultations in the 2019 post-period, as compared to 358 in the 2020 post-period). By comparison, the total number of ED arrival volumes decreased by 28.8% (20,198 total ED presentations in the 2019 post-period, as compared to 14,358 in the 2020 post-period).

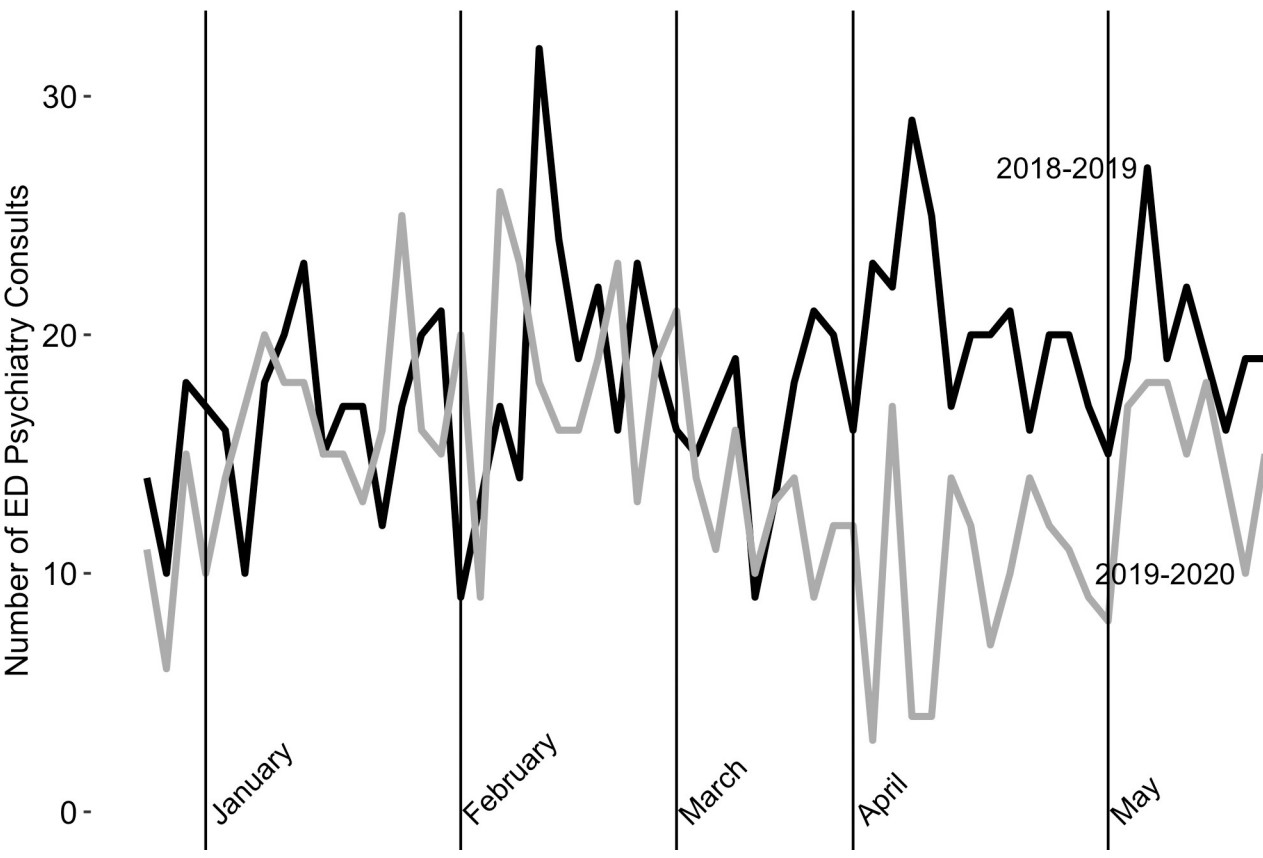

**Fig 2. Comparison of unadjusted trends in psychiatry consults between 2019 and 2020 cohorts.**

In exploring pre-period differences, we found that the COVID-19 pre-period (December 23, 2019–February 24, 2020) cohort had a higher proportion of psychiatric ED consultations involving people from a minority racial group than did the comparator pre-period (December 24, 2018–February 25, 2019) (Table 1). We further explore these differences in S1 Table by presenting an unadjusted comparison of racial/ethnic groups stratified by psychiatric presentation in the pre-period. No statistically significant pre-period differences were found when comparing group variables including sex, median age, and insurance payer.

We found that rates of psychiatric presentation were similar between the two study groups in the pre-period for patients presenting with suicidal ideation, SUDs, and affective disorders (Table 2). We found that fewer patients with psychotic disorders were presenting in the 2019–2020 pre-period cohort, as compared with the 2018–2019 pre-period cohort ($p = 0.04$).

Our adjusted models for each of the four psychiatric presentation groups (suicidal ideation, affective disorders, psychotic disorders, and SUDs) for the two study periods, which run from December to May in 2018–2019 and 2019–2020, respectively, are summarized in Fig 3, with tabular data presented in S2 Table. These results suggest that after the COVID-19 pandemic began spreading in Boston, there was no significant differential change in the proportion of ED presentations for affective disorders or psychotic disorders at any point in the COVID-19 time series relative to the comparator time series.

In the adjusted models, we found differential changes in the model proportions for both presentations with suicidal ideation, as well as SUDs. For psychiatric consultations with suicidal ideation present, we found a significant differential increase six weeks after the exposure,

**Table 1. Pre-period unadjusted comparison of demographic characteristics.**

| Characteristic | 2018–2019 Cohort (n = 489) | 2019–2020 Cohort (n = 467) | Test-statistic (*p*-value) |
|---|---|---|---|
| Age, median, years | 38 | 39 | – |
| Sex, n (%) | | | |
| Male | 281 (57.5) | 301 (58.0) | 0.2 (0.9) |
| Female | 208 (42.5) | 196 (42.0) | |
| Racial Classification, n (%) | | | |
| White | 383 (78.3) | 320 (68.5) | 3.4 (<0.001) |
| Black | 55 (11.2) | 79 (16.9) | -2.5; 0.01 |
| Asian | 9 (1.8) | 17 (3.6) | -1.7; 0.09 |
| American Indian/Alaska Native | 5 (1.0) | 2 (0.4) | 1.1; 0.3 |
| Other | 32 (6.5) | 38 (8.1) | -0.9; 0.02 |
| Hispanic or Latino/a/x, n (%) | | | |
| Yes | 47 (9.9) | 53 (11.9) | -1.0; 0.20 |
| No | 426 (80.1) | 391 (78.1) | |
| Insurance, n(%) | | | |
| MA Medicaid | 194 (29.7) | 177 (37.9) | 0.6; 0.6 |
| Private Insurance | 140 (28.6) | 136 (29.1) | -0.2; 0.9 |
| Medicare | 92 (18.8) | 107 (22.9) | -1.6; 0.1 |
| Uninsured | 24 (.9) | 13 (2.8) | 1.7; 0.09 |
| Other public coverage | 39 (8.0) | 34 (7.3) | 0.4; 0.7 |

or the week of April 13–17, 2020 (36.4 percentage points change; 95% CI: 5.3, 67.6). Based on the comparison series, we would have expected an increase of 16.0 percentage points for suicidal ideation at week six. Relative to what we would have expected based on presentations from the previous year during week six, we observed a 127.5% increase in presentations with suicidal ideation following the coronavirus outbreak. Additionally, at nine weeks after the exposure (May 4–9, 2020), there was a differential decrease in the proportion of psychiatric presentations to the ED with suicidal ideation (34.9 percentage points; 95% CI; -69.5, -0.3), though this change was not statistically significant (*p* = 0.06). We calculated the percent increase for suicidal ideation presentations (as well as for SUD presentations below) by dividing the differential change in the post-period (i.e., the treatment effect estimate) by the difference in means from the pre-period to the post-period in the comparison group (i.e., the counterfactual).

We present data in S3 Table for suicidal ideation, sequentially removing and re-adding other certain psychiatric presentations to the model. Our results change modestly because of this different regression specification. There is a significant increase in presentations with suicidal ideation at weeks three and six. However, once adjustment for SUD is included in the model at week three, the differential change is no longer significant because this increase was mediated by the co-occurring SUD diagnoses. However, even after the adjustments for co-

**Table 2. Pre-period unadjusted comparison of psychiatric presentations.**

| Psychiatric Presentation n (%) | 2018–2019 Cohort (n = 489) | 2019–2020 Cohort (n = 467) | Test statistic; *p*-value |
|---|---|---|---|
| Suicidal Ideation | 293 (59.9) | 274 (58.7) | 0.4; 0.7 |
| Substance Use Disorder | 202 (41.3) | 184 (39.4) | 0.6; 0.5 |
| Affective Disorder | 172 (35.2) | 184 (39.4) | -1.35; 0.2 |
| Psychotic Disorder | 83 (17.0) | 57 (12.2) | 2.1; 0.04 |

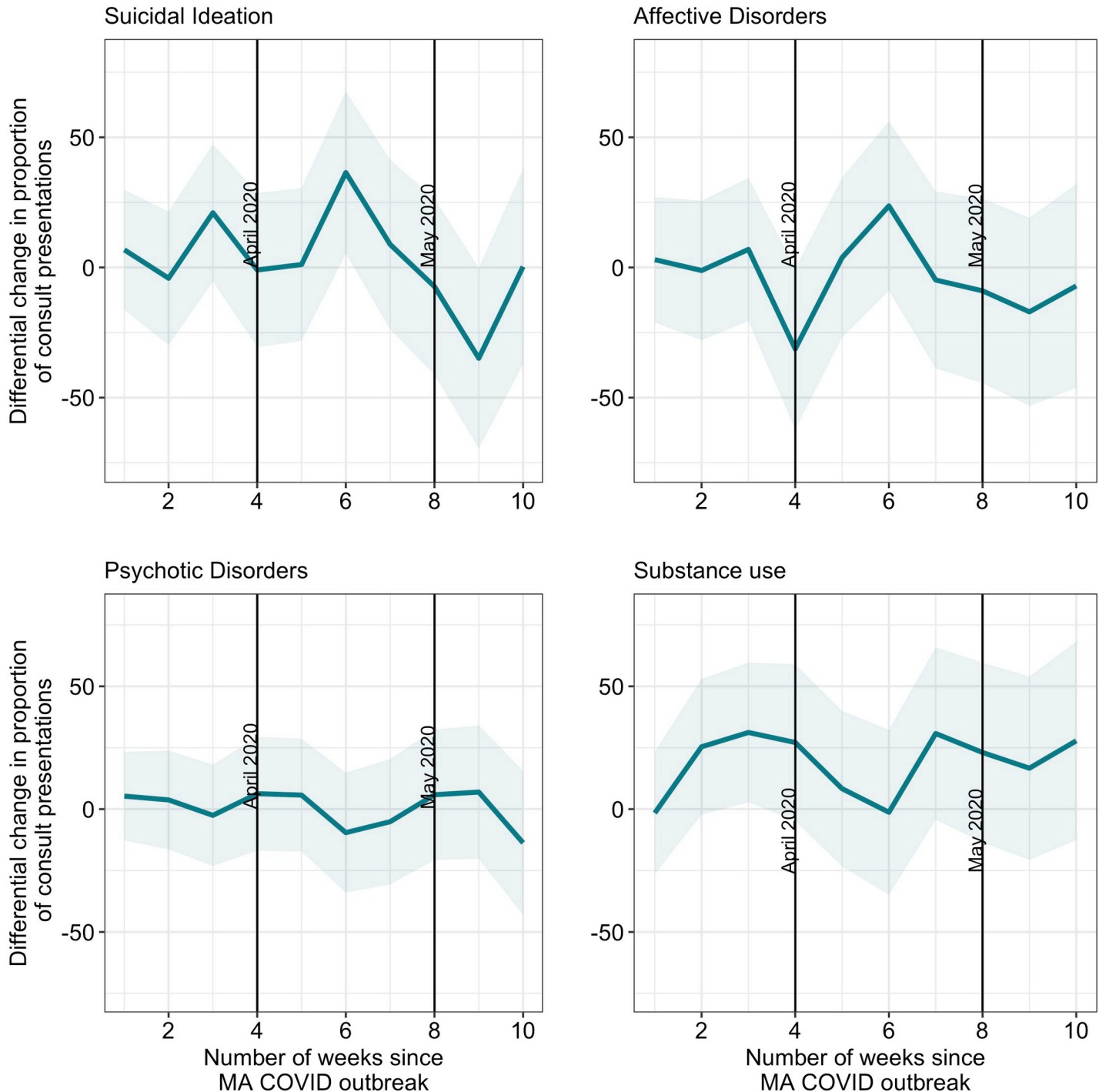

**Fig 3. Differential changes in ED psychiatry consult presentations in the COVID versus non-COVID post periods.**

occurring psychiatric conditions are added at week six, the differential change persists, as noted above.

There is also a differential increase in the adjusted proportion of presentations with SUD in the COVID-19 time series relative to the comparison time series at all time points after the exposure. This differential increase is statistically significant at three weeks following the

exposure, or the week of March 23–27, 2020 (32.8 percentage points; 95% CI: 4.0, 61.6). Based on the comparison series, we would have expected a decrease of 6.2 percentage points for substance use at week three. Relative to what we would have expected based on diagnoses from the previous year during week three, we saw a 629.0% increase in SUD presentations following the coronavirus outbreak.

## Discussion

While overall ED psychiatric presentations decreased during the COVID-19 pandemic in Boston, the proportion of psychiatric presentations with suicidal ideation and SUDs increased.

The proportion of presentations with suicidal ideation increased six weeks after the exposure (April 13–17, 2020) or approximately three weeks after the March 23, 2020 shelter-in-place order. This was also the week leading up to the peak of the initial surge of COVID-19 cases in Massachusetts (4,946 new cases reported on April 24, 2020) [31]. This increase in the proportion of psychiatric presentations with suicidal ideation may reflect increasing stress as a result of the lockdown and isolation [1, 2]. The influence of having COVID, knowing people who had COVID, or fears about being infected with the coronavirus may also be related to this finding. While another recent Massachusetts-based study suggests that overall suicide rates did not increase during the pandemic, it appears that the experience of suicidal ideation may have increased at certain time points following the outbreak [13]. This finding regarding increased pandemic-related suicidal ideation, as well as our finding regarding increased presentations with SUD, is corroborated by another recent large-scale cross-sectional study in the United States [32].

Of note, in the adjusted analyses, at nine weeks after the exposure (May 4–9, 2020), there was a meaningful differential decrease in the proportion of psychiatric presentations to the ED with suicidal ideation. This decrease in suicidal ideation presentations parallels Governor Charles Baker's announcement of the Reopening Advisory Board (April 28, 2020) [33] and the subsequent four-phase reopening plan announcement on May 11, 2020 [34]. Additionally, the month of May brought to Boston weather with an increased average daily temperature compared to the previous month, potentially facilitating outdoor socializing and connection. Unlike the trends noted in SUDs presentations, ED psychiatric presentations with suicidal ideation were dynamic across the study period.

Trends in presentations for SUDs suggest that these illnesses may have worsened during the pandemic, consistent with expert predictions and other findings [11, 21, 35, 36]. Some patients may have escalated substance use as a means of coping with the stress of the pandemic and shutdown [11]. Additionally, SUDs are highly prevalent in populations experiencing houselessness [37]. Given the higher risk of COVID-19 in communities without stable housing, several shelters in Boston changed their access policies to mitigate the spread of disease [35, 38]. Though important for reducing viral spread, these policies potentially resulted in housing barriers for communities experiencing houselessness, which may have disproportionately affected persons with SUDs [35, 38]. Furthermore, individuals with SUDs may have faced challenges accessing substances during quarantine, leading to increased presentations for withdrawal-related symptoms. Additionally, patients may have struggled accessing treatment, as detoxification and residential care programs were initially limiting access to patients without a known COVID-19 status and not accepting patients who tested positive for COVID, even if asymptomatic. Most concerning, perhaps, are cases involving individuals who experienced decreased tolerance to substances in the setting of weeks of reduced access and thus were more likely to unintentionally overdose during subsequent substance use [36]. Additionally, our sensitivity analyses suggest that SUD presentations seem to explain the increase in suicidal

ideation presentations in week 3 in the unadjusted models presented in S3 Table, given that estimates become non-significant once SUD was included as an adjustment. This aligns with our findings that there is a significant increase in SUD presentations following the COVID-19 outbreak.

We found pre-period differences between the 2018–2019 and 2019–2020 cohorts, with a higher proportion of psychiatric ED consultations involving people from a minority racial group in the latter 2019–2020 cohort. When we further explore this difference by separating the groups by psychiatric presentations, statistically significant differences are isolated to fewer White patients presenting with psychotic disorders and more Asian patients presenting with affective disorders in the 2019–2020 cohort. The reason behind these pre-period differences remains unclear, though perhaps ED staff training regarding documentation improved the accuracy of recording data on race and ethnicity for the 2019–2020 cohort.

A better understanding of overall mental health resource utilization during the pandemic is needed, particularly in the setting of decreased psychiatric ED presentations following the outbreak, as well as documented expansion of telehealth services [39]. Patients in need of psychiatric care may have worried about potential risk for coronavirus infection associated with ED presentation and may have chosen to access treatment through other modalities, such as telehealth services, crisis support lines, and employee assistance programs. Another grimmer possibility is that patients went without treatment for weeks until access improved or until their symptoms became so burdensome that they could no longer be ignored [40]. Given the dynamic trends in suicidal ideation observed during the study period, further work is needed to better characterize phenotypes of patients presenting with suicidal ideation during the pandemic, including the triggers for suicidal ideation, underlying co-morbidities, and the presence of plan, intent, and attempts. Additionally, identification of patient populations vulnerable to suicidal ideation and attempt is requisite to appropriately target interventions.

Moving forward, guaranteed access to community-based mental health care and addiction services will be important [11, 12]. Specifically, maintenance of accessibility to resources, as well as insurance coverage for telehealth services, is prudent. We recommend raising awareness surrounding the potential deleterious relationship between the pandemic and mental illness, SUDs, and suicidal ideation.

## Limitations

Our study has limitations. First, circumstances unique to Boston may have contributed to our findings, including those that we could not identify, potentially resulting in less generalizability outside of this area. Second, our study did not capture suicidal ideation, intent, plan, and attempt in patients who did not present for care. Third, we stopped collecting data in May, consistent with the end of the initial surge in Massachusetts. Because of this data limitation, we do not capture trends of psychiatric presentations that occurred after the surge, which may still have been related to the initial outbreak. Fourth, in our generation of outcome variables, we relied on clinician reviewer interpretation of consultation notes, which also represent interpretations of clinical presentations. We hope that the reviewers added a second layer of assessment, thus improving diagnostic clarity, though understand that the additional review may have introduced reviewer bias. Fifth, while the confidence intervals for some of our results are large, they generally do not suggest that we are underpowered to detect an effect (i.e., an unstably estimated zero) except for presentations with psychosis. In most cases, we can rule out large estimates in the opposite direction of the treatment effect, suggesting that the estimated treatment effects may be under-estimates of the increases or decreases seen in our study. Additionally, with a general CITS, estimates further from the exposure typically have larger

confidence intervals, as uncertainty increases over the time period when estimating a between-group difference from the data. Finally, there were many interventions to address the spread of COVID-19 during our study period. While we use a robust quasi-experimental design to capture changes in psychiatric presentations to ED unrelated to COVID-19, our pattern of findings may be attributed to more than the policy changes cited. Thus, caution is warranted when interpreting our results as causal.

## Conclusions

The COVID-19 pandemic is associated with changes in ED presentations requiring psychiatric consultation, particularly those involving suicidal ideation and SUDs.

## Supporting information

**S1 Material.**
(DOCX)

**S1 Table. Pre-period unadjusted comparison of racial/ethnic groups by psychiatric presentation.**
(DOCX)

**S2 Table. Full regression output for differential changes in emergency department psychiatric presentations in the 2019 versus 2020 post-periods.**
(DOCX)

**S3 Table. Differential change in suicidal ideation presentations between COVID-19 time series (Dec 2019–May 2020) and comparator time series (Dec 2018–May 2019) from pre-period (Dec–Feb) to post-period (Mar–May) among all ED psychiatric presentations, varying adjustment for other co-occurring psychiatric conditions.**
(DOCX)

## Acknowledgments

All contributors to this manuscript met criteria for authorship.

## Author Contributions

**Conceptualization:** Michal J. McDowell, Mila Grossman, Charles Masaki, Emily Sorg, Scott R. Beach.

**Data curation:** Michal J. McDowell, Mladen Nisavic, Mila Grossman, Charles Masaki, Emily Sorg, Scott R. Beach.

**Formal analysis:** Michal J. McDowell, Carrie E. Fry, Mladen Nisavic, Mila Grossman, Charles Masaki, Emily Sorg, Scott R. Beach.

**Funding acquisition:** Michal J. McDowell.

**Investigation:** Michal J. McDowell, Mladen Nisavic, Scott R. Beach.

**Methodology:** Michal J. McDowell, Carrie E. Fry, Mila Grossman, Charles Masaki, Emily Sorg, Scott R. Beach.

**Project administration:** Michal J. McDowell, Mladen Nisavic, Suzanne Bird, Felicia Smith, Scott R. Beach.

**Resources:** Michal J. McDowell, Suzanne Bird, Felicia Smith, Scott R. Beach.

**Software:** Carrie E. Fry.

**Supervision:** Scott R. Beach.

**Validation:** Michal J. McDowell, Carrie E. Fry, Scott R. Beach.

**Visualization:** Carrie E. Fry, Scott R. Beach.

**Writing – original draft:** Michal J. McDowell, Carrie E. Fry, Mila Grossman, Scott R. Beach.

**Writing – review & editing:** Michal J. McDowell, Carrie E. Fry, Mladen Nisavic, Mila Grossman, Charles Masaki, Emily Sorg, Suzanne Bird, Felicia Smith, Scott R. Beach.

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
