## [Decision Letter · Decision Letter 0]

18 Mar 2021

PONE-D-21-02006

Evaluating the Association Between COVID-19 and Psychiatric Presentations, Suicidal Ideation in an Emergency Department: An Interrupted Time Series Study

PLOS ONE

Dear Dr. McDowell,

Thank you for submitting your manuscript to PLOS ONE. After careful consideration, we feel that it has merit but does not fully meet PLOS ONE’s publication criteria as it currently stands. Therefore, we invite you to submit a revised version of the manuscript that addresses the points raised during the review process.

We look forward to receiving your revised manuscript.

Kind regards,

Corstiaan den Uil

Academic Editor

PLOS ONE

Journal Requirements:

"- Mass General Brigham

- #2020P001482

- The IRB has determined that this project meets the criteria for exemption 45 CFR

46.101(b)(#)

- Consent was not obtained".   

3. Please provide additional details regarding participant consent.

If you are reporting a retrospective study of medical records or archived samples, please ensure that you have discussed whether all data were fully anonymized before you accessed them and/or whether the IRB or ethics committee waived the requirement for informed consent.

For additional information about PLOS ONE ethical requirements for human subjects research, please refer to " ext-link-type="uri" xlink:type="simple">http://journals.plos.org/plosone/s/submission-guidelines#loc-human-subjects-research."

4. n your Methods section, please ensure you have provided sufficient details to replicate the analyses such as  a description of any inclusion/exclusion criteria that were applied to participant inclusion in the analysis.

Moreover, please specify the name of the centre where the research took place.

5. Please refer to the specific statistical analyses performed as well as any post-hoc corrections to correct for multiple comparisons. If these were not performed please justify the reasons.

Please refer to our statistical reporting guidelines for assistance (https://journals.plos.org/plosone/s/submission-guidelines.#loc-statistical-reporting).

6. We note that you have indicated that data from this study are available upon request. PLOS only allows data to be available upon request if there are legal or ethical restrictions on sharing data publicly. For information on unacceptable data access restrictions, please see http://journals.plos.org/plosone/s/data-availability#loc-unacceptable-data-access-restrictions.

Reviewers' comments:

Reviewer's Responses to Questions

**Comments to the Author**

1. Is the manuscript technically sound, and do the data support the conclusions?

Reviewer #1: Yes

2. Has the statistical analysis been performed appropriately and rigorously? 

Reviewer #1: Yes

3. Have the authors made all data underlying the findings in their manuscript fully available?

Reviewer #1: Yes

4. Is the manuscript presented in an intelligible fashion and written in standard English?

Reviewer #1: Yes

5. Review Comments to the Author

Reviewer #1: This study examines changes in the prevalence of suicidality and selected psychiatric disorders in a busy ED during the early phase of the COVID-19 pandemic, relative to earlier time periods. The investigation addresses a concern of obvious interest and relevance. Several studies have reported changes in psychiatric case rates in various settings, but very few within a US ED. The paper is well written.

The investigators might consider some of the following comments to further enhance the manuscript:

1. In the study aims section (p. 4, last paragraph of the Introduction), it may be helpful to specify the study outcomes (even if only in parentheses) —“psychiatric presentations” is somewhat vague.

2. Throughout the Methods and Results, it might be useful to clarify the nature of the dependent variables. Did the analyses examine changes in the PERCENTAGES of psychiatric cases in the ED presenting with a given psychiatric diagnosis or SI, (e.g., the percentage of psychiatric cases with an affective disorder), or the raw NUMBER of these cases? Similarly, it may be helpful to indicate more explicitly for readers whether the repeated reference to “rates” signifies percentages. The same basic point applies to the Figures.

3. In the Outcome Measures section (p. 7), it may be helpful to define the outcomes more explicitly. For example, although the text on line 150 refers to “suicidality,” line 153 refers more specifically to “suicidal ideation.” More importantly, it would be useful to indicate how reports of previous as opposed to current SI were handled (presumably these were not counted); how intermittent use of substances were differentiated from diagnosed substance use disorders; and whether or how previous psychiatric disorders were differentiated from current ones. (In other words, was a patient who presented with a prior hx of recurrent major depression in full remission counted the same as one presenting with an active depressive disorder?).

4. In the Methods section (p. 7), it may be helpful to offer a brief rationale for the diagnoses selected as the outcomes for this study (e.g., why were mood disorders analyzed but not anxiety disorders?).

5. The regression model regarding suicidal ideation adjusted for various psychiatric disorders— it would be interesting as well to see the findings without these adjustments. Similarly, the analyses of psychiatric disorders adjusted for selected demographic variables and total number of psychiatric consultations; it might be useful to note the results without these adjustments (but of course retaining the time-related covariates).

6. The Results section (p. 8, line 175) reports that the overall volume of psychiatric consultations declined in the COVID-19 post period compared with the comparator post-period. Did the number of non-psychiatric consultations decline significantly as well? This information would provide readers with greater context for understanding the change in number of psychiatric consultations.

7. The Results section (p. 9, lines 190-191) indicates that “the rates of psychiatric presentation were similar between the two study groups in the pre-period.” However, Table 2 appears to indicate a significant change in the number of psychotic disorders; if so, it would be useful to mention this in the text.

8. In the Results section (p. 12, 1st paragraph), I confess that I had some difficulty following the calculation of percent increases (e.g., 30.7% vs. 558%). Perhaps the investigators might wish to expand the explanation a bit.

9 In the Discussion section, the authors might consider integrating their findings with other studies regarding changes in volume of psychiatric cases associated with the COVID-19 pandemic.

10. The study limitations section (pp. 15-16) makes some good observations. However, the large confidence intervals receive only fleeting reference, which does not really seem to do justice to this issue— it may be helpful to address this concern a bit more fully, including the ramifications of the very large CI’s for interpreting the findings.

11. Table 2 refers to “mean (%)” in the first column, but evidently only percentages are reported— it would be useful to include the frequencies as well, and to specify how the percentages were calculated (percentage of total number of psychiatric cases during that interval, percentage of total [psychiatric and non-psychiatric] ED visits during that interval, etc.?).

Table 1 reports percentages. It would be conventional to report the frequencies as well (though I appreciate that journals vary in their preferences and the editor may have a different view).

I was unable to find Appendix in the material I reviewed.

MINOR comments:

1. The use of the term “intervention” throughout the paper may be confusing to readers, since there is no intervention in the traditional clinical sense. Perhaps the authors might consider “exposure” instead.

2. On p. 7 (lines 146-147), the text states that information was extracted “automatically” for some variables and “manually” for others. It is not entirely clear what this means (e.g., the medical records could be queried for summary information regarding some variables, but information had to be manually extracted by searching each record individually for other variables?).

3. On p. 11 (lines 196-197), the text notes that Figure 3 summarizes the “adjusted models for each of the four psychiatric presentation groups.” It may be useful to specify the time period for these data (e.g., evidently the entire study period from Dec. to May).

4. On p. 11, in the last paragraph regarding changes in SI and SUDS, it would be helpful to reiterate that these analyses were adjusted (this was mentioned in the previous paragraph but not in this one).

5. On p. 3, line 57, it would be useful to note that the 30% in suicide rates associated with the 2003 SARS epidemic was reported in a study conducted in Hong Kong, if that is accurate (the previous clause refers to research in the US). Similarly, on p. 12 (lines 236), the last line on the page summarizes findings from the Faust et al. study-- it may be useful to specify that this investigation focused on rates in Massachusetts.

6. PLOS authors have the option to publish the peer review history of their article (what does this mean?). If published, this will include your full peer review and any attached files.

Reviewer #1: No

---

## [Author Response · Author response to Decision Letter 0]

30 Apr 2021

Manuscript: PONE-D-21-02006

Title: “Evaluating the Association Between COVID-19 and Psychiatric Presentations, Suicidal Ideation in an Emergency Department: A Comparative Interrupted Time Series Study”

Response to Reviewers

Dear Editor:

We thank you for the thoughtful feedback on our manuscript. In the letter below, we outline how we have addressed the reviewers’ questions and concerns for our submission to PLOS ONE. We have additionally edited the manuscript to ensure that it follows guidelines outlined in your letter.

Again, thank you for your consideration of our manuscript.

Respectfully, 

Michal J. McDowell, MD MPH

Journal Requirements:

Thank you for these helpful references. We have amended our article title, subtitle, and short title to be written in sentence case. We have amended our Level 2 headings to use sentence case, as well.

"- Mass General Brigham

 - #2020P001482

 - The IRB has determined that this project meets the criteria for exemption 45 CFR

 46.101(b)(#)

 - Consent was not obtained". 

Thank you for this helpful reference. We have amended our ethics statement to include the full name of the institutional review board that approved our study. Additionally, we have added this statement to the first paragraph of the Methods section of the manuscript: “The Partners Institutional Review Board approved this work (approval #2020P001482) and waived the requirement for informed consent.”

3. Please provide additional details regarding participant consent.

If you are reporting a retrospective study of medical records or archived samples, please ensure that you have discussed whether all data were fully anonymized before you accessed them and/or whether the IRB or ethics committee waived the requirement for informed consent.

For additional information about PLOS ONE ethical requirements for human subjects research, please refer to http://journals.plos.org/plosone/s/submission-guidelines#loc-human-subjects-research."

Thank you for this helpful reference. We have amended our ethics statement to specify that the institutional review waived the requirement for informed consent. Additionally, we have added this statement to the first paragraph of the Methods section of the manuscript: “The Partners Institutional Review Board approved this work (approval #2020P001482) and waived the requirement for informed consent.”

4. n your Methods section, please ensure you have provided sufficient details to replicate the analyses such as a description of any inclusion/exclusion criteria that were applied to participant inclusion in the analysis.

Moreover, please specify the name of the centre where the research took place.

Thank you for this amendment. In the first paragraph of our methods section, we now specify the center at which the research took place, Massachusetts General Hospital. We also now clarify our inclusion criteria on page 6 of our Methods section, “Patients were included if they received a psychiatric consultation in any one of the four observation periods (pre and post-period for comparator time series or pre and post-period for COVID time series). Patients were allowed to appear in more than one of these periods. Patients that did not receive a psychiatric consultation (even if one was requested) were excluded from this study. Additionally, patients with incomplete demographic or diagnostic records were excluded from analyses that used the characteristic of interest, including CITS regression modeling.”

5. Please refer to the specific statistical analyses performed as well as any post-hoc corrections to correct for multiple comparisons. If these were not performed please justify the reasons.

Please refer to our statistical reporting guidelines for assistance (https://journals.plos.org/plosone/s/submission-guidelines.#loc-statistical-reporting).

Thank you for this helpful reference. 

Regarding statistical details, we now specify the following:

• Our statistical testing was executed between subjects, as we now indicate in the Study design sub-section of the Methods section. 

• Regarding assumptions, we now make further clarifications about our assumptions and checks surrounding linearity and normality. 

o Regarding linearity of the outcomes over the study period, we now clarify our assumption of linearity at the end of the second paragraph of the Study design sub-section of our Methods section: “A general CITS assumes that any differential growth between the two groups is linear, and we conducted an event study to visually assess this assumption.” 

o Regarding normality, we assume that the distribution of our outcomes is normal and that the relationship between the outcome and predictors is linear and additive. Typically, researchers use a non-linear regression model (e.g., logistic regression) when the outcome is binary, which assumes that the relationship between the predictors and outcome is multiplicative. We chose to use linear regression, as opposed to logistic regression to facilitate easy interpretation of the treatment effect estimates, which are interactions. In the first full paragraph of the Data analysis sub-section of our Methods section, we now give further background on how and why we estimated the linear probability models: “LPMs are preferred to logit or probit regression models with binary outcomes for both CITS and difference-in-differences, due to the ease of interpreting interactions in LPMs relative to logit or probit models.”

• There were no explicit outliers in our data. Regarding how we handled missing data, we now clarify that observations with missing data were excluded (fifth paragraph of the Study design sub-section of our Methods section): “Patients that did not receive a psychiatric consultation (even if one was requested) were excluded from this study. Additionally, patients with incomplete demographic or diagnostic records were excluded from analyses that used the characteristic of interest, including CITS regression modeling.”

• We also now clarify that we adjust for multiple comparisons testing with Bonferroni corrections (see paragraph 3 of the Data analysis sub-section of our Methods section). 

• For our statistical tests, we now add descriptions our statistical tests in the first paragraph of the Data analysis sub-section of the Methods section: “First, we compared demographic and clinical characteristics among the pre-period (December – March) of each time series using a two-sided t-test to measure baseline similarity between these two groups. Then, we estimated linear probability models (ƒ-lines) for each of our four outcomes.”

Additionally, we now include the full regression output in the supplemental materials (see Appendix Table 2). 

Regarding p-values, we now ensure that exact values are reported. We now provide both test statistics and p-values for each test.

Regarding percentages, we now ensure that we provide the numerator and denominator for all percentages. 

6. We note that you have indicated that data from this study are available upon request. PLOS only allows data to be available upon request if there are legal or ethical restrictions on sharing data publicly. For information on unacceptable data access restrictions, please see http://journals.plos.org/plosone/s/data-availability#loc-unacceptable-data-access-restrictions.

Thank you for this helpful reference. 

We have revised our cover letter to address the prompts indicated: “There are ethical restrictions on sharing our data set, as our data contains potentially identifying and sensitive patient information per the Partners Institutional Review Board. Data is available upon request. Requests may be sent to the corresponding author or to the Partners Institutional Review Board protocol administrator, Virginia Rodriguez VGRODRIGUEZ@PARTNERS.ORG. For reference, our protocol number for the review board is 2020P—1482.”

Reviewers' comments:

Reviewer's Responses to Questions

Comments to the Author

1. Is the manuscript technically sound, and do the data support the conclusions?

Reviewer #1: Yes

Thank you for the support regarding the technical rigor of the manuscript. 

2. Has the statistical analysis been performed appropriately and rigorously? 

Reviewer #1: Yes

Thank you for the support regarding the statistical analysis performance.

3. Have the authors made all data underlying the findings in their manuscript fully available?

Reviewer #1: Yes

Thank you, we agree that the data presented in the findings is available.

4. Is the manuscript presented in an intelligible fashion and written in standard English?

Reviewer #1: Yes

Thank you for the endorsement of the manuscript presentation.

5. Review Comments to the Author

Reviewer #1: This study examines changes in the prevalence of suicidality and selected psychiatric disorders in a busy ED during the early phase of the COVID-19 pandemic, relative to earlier time periods. The investigation addresses a concern of obvious interest and relevance. Several studies have reported changes in psychiatric case rates in various settings, but very few within a US ED. The paper is well written.

Many thanks for the praise and kind feedback.

The investigators might consider some of the following comments to further enhance the manuscript:

1. In the study aims section (p. 4, last paragraph of the Introduction), it may be helpful to specify the study outcomes (even if only in parentheses) —“psychiatric presentations” is somewhat vague.

Thank you for this thoughtful feedback. We have modified from “psychiatric presentations” to “patients presenting with psychiatric complaints” in the last paragraph of the Introduction section. We also further clarify this point in the first paragraph of the Study design sub-section of the Methods section: “We used an interrupted time series design to compare changes in the proportion of patients seen with certain psychiatric presentations before and after the COVID-19 outbreak.”

2. Throughout the Methods and Results, it might be useful to clarify the nature of the dependent variables. Did the analyses examine changes in the PERCENTAGES of psychiatric cases in the ED presenting with a given psychiatric diagnosis or SI, (e.g., the percentage of psychiatric cases with an affective disorder), or the raw NUMBER of these cases? Similarly, it may be helpful to indicate more explicitly for readers whether the repeated reference to “rates” signifies percentages. The same basic point applies to the Figures.

Thank you for this helpful suggestion. We have taken steps to clarify the nature of the dependent variables. 

Our analyses examined changes in the percentages of psychiatric cases in the ED. Specifically, the numerator for our proportion is patients presenting with a given psychiatric diagnosis and the denominator is all patients seen by the psychiatric consult service in the ED. 

We now clarify this point in the Outcome measures sub-section in the Methods section: “Our outcomes of interest were the proportion of patients who presented with suicidal ideation (SI), SUDs (including alcohol, opioid, cocaine, and stimulant use disorders, as well as substance-induced mood and psychotic disorders), psychotic disorders (including schizophrenia-spectrum illnesses and brief psychotic disorder), and affective disorders (including major depressive disorder, and bipolar affective disorder) among all patients seen by the psychiatric consult service.” 

We also agree that indicating explicitly for readers that the rates signify percentages is important. We have changed the language throughout the document from “rate” to “proportion” to clarify. 

Additionally, we have updated Fig 2 to add a clarification that the Y-axis is number based. We have updated Fig 3 to clarify that we indicate differential changes in proportion. 

3. In the Outcome Measures section (p. 7), it may be helpful to define the outcomes more explicitly. For example, although the text on line 150 refers to “suicidality,” line 153 refers more specifically to “suicidal ideation.” More importantly, it would be useful to indicate how reports of previous as opposed to current SI were handled (presumably these were not counted); how intermittent use of substances were differentiated from diagnosed substance use disorders; and whether or how previous psychiatric disorders were differentiated from current ones. (In other words, was a patient who presented with a prior hx of recurrent major depression in full remission counted the same as one presenting with an active depressive disorder?).

Thank you for this comment, which we believe helps clarify our outcome measures.

We agree that “suicidality” is not clear, so we have changed this term to “suicidal ideation” throughout the manuscript. 

We agree that clarification surrounding various outcome variables is necessary. We now touch on how we handled past versus present suicidal ideation, substance use versus substance use disorder, and past versus present psychiatric diagnoses in the Outcome measures sub-section of the Methods section: “We only included present suicidal ideation in our outcome variable. For psychiatric diagnoses, we generally used the diagnosis given by the evaluating clinician. Historical labels were only used if they were deemed correct by the evaluating clinician and relevant to the current presentation. For SUDs, historical diagnoses were used if substances were involved in the current presentation or if there was compelling evidence of an SUD in the consult note.”

4. In the Methods section (p. 7), it may be helpful to offer a brief rationale for the diagnoses selected as the outcomes for this study (e.g., why were mood disorders analyzed but not anxiety disorders?).

Thank you for this thoughtful suggestion – we agree that offering a rationale for the diagnoses we selected as the outcomes for this study would be useful for readers.

We now make this clarification in the Outcome measures sub-section of the Methods section: “We included suicidal ideation, SUDs, psychotic disorders, and affective disorders as our outcomes of interest because, based on our clinical experience, these diagnoses are reliably and consistently captured in our ED setting. Anxiety disorders, although coded by the reviewers, represent a small proportion of ED visits and unless they are the primary reason for presentation, they are more likely to be inconsistently captured.”

We also added further clarification regarding the means by which reviewers identified diagnoses in this sub-section: “Reviewers included relevant and current diagnoses identified in the consult notes, as well as interpreted symptoms mentioned with Diagnostic and Statistical Manual of Mental Disorders 5th edition (DSM-5) criteria in mind.”

Additionally, we included an addition to our limitation section regarding potential bias in reviewer interpretation of clinical notes, and thus outcome variables: “Fourth, in our generation of outcome variables, we relied on clinician reviewer interpretation of consultation notes, which are also represent interpretations of clinical presentations. We hope that the reviewers added a second layer of assessment, thus improving diagnostic clarity, though understand that the additional review may have introduced reviewer bias.”

We have also removed the bottom three rows of Table 2 (which detailed personality disorder, PTSD, and generalized anxiety disorder) to streamline outcome measure reporting.

5. The regression model regarding suicidal ideation adjusted for various psychiatric disorders— it would be interesting as well to see the findings without these adjustments. Similarly, the analyses of psychiatric disorders adjusted for selected demographic variables and total number of psychiatric consultations; it might be useful to note the results without these adjustments (but of course retaining the time-related covariates).

Thank you for this thoughtful suggestion.

We agree that unadjusted analyses are often useful to explore.

We now include the regression model for suicidal ideation presentations for various psychiatric disorders with varying adjustments in Appendix Table 3. We now clarify this inclusion in the Data analysis sub-section of the Methods section: “For suicidal ideation, we also adjusted for the presence of an affective disorder, psychotic disorder, and SUD. As a sensitivity analysis, we estimate the same regression for suicidal ideation in Appendix Table 3, omitting all of the other diagnostic categories as covariates and adding back one of the diagnostic categories, since they may be on the causal pathway between exposure and suicidal ideation. Our results change modestly as a result of this different regression specification.”

We do not report findings without demographic adjustments because the demographic composition of cohorts may be changing differentially from pre- to post-period. We agree that clarifying why we do not present this analysis would be useful for the reader. We now include a sentence to this end in the Data analysis sub-section of the Methods section: “We do not present data not adjusted by demographic factors, as the demographic composition of cohorts may be change in different ways in the pre- to post-period.”

Of note, we corrected an error in Table 1 that we identified in reporting unadjusted comparisons of ethnicity in the pre-period. We corrected data related to this error throughout the manuscript.

6. The Results section (p. 8, line 175) reports that the overall volume of psychiatric consultations declined in the COVID-19 post period compared with the comparator post-period. Did the number of non-psychiatric consultations decline significantly as well? This information would provide readers with greater context for understanding the change in number of psychiatric consultations.

Thank you for this excellent addition – we agree that understanding of total consultations would be helpful to understand as well.

We reached out to the EHR data team for data regarding total ED volumes, which we now include in the first paragraph of our Results section.

7. The Results section (p. 9, lines 190-191) indicates that “the rates of psychiatric presentation were similar between the two study groups in the pre-period.”

 However, Table 2 appears to indicate a significant change in the number of psychotic disorders; if so, it would be useful to mention this in the text.

Thank you for highlighting this. We now have separated the paragraph in the Results section discussing Table 2. We draw the reader’s attention to the significant difference in psychotic disorder presentations in the pre-period for the 2018-2019 cohort as compared to the 2019-2020 cohort: “We found that fewer patients with psychotic disorders were presenting in the 2019-2020 pre-period cohort, as compared with the 2018-2019 pre-period cohort (p = 0.04).”

8. In the Results section (p. 12, 1st paragraph), I confess that I had some difficulty following the calculation of percent increases (e.g., 30.7% vs. 558%). Perhaps the investigators might wish to expand the explanation a bit.

We agree that this was difficult to follow as written – thank you for highlighting. We have changed the wording in the two paragraphs following Figure 3 in the Results section and hope the language is now more accessible: 

“We found differential changes in the adjusted proportion of presentations with suicidal ideation and SUDs. We also present unadjusted data in Appendix Table 3 for suicidal ideation. For psychiatric consultations with suicidal ideation present, we found a significant, differential increase six weeks after the exposure, or April 13-17, 2020 (36.4 percentage points change; 95% CI: 5.3, 67.6). Based on the comparison series, we would have expected an increase of 16.0 percentage points for suicidal ideation at week six. Relative to what we would have expected based on presentations from the previous year during week six, we observed a 127.5% increase in presentations with suicidal ideation following the coronavirus outbreak. We calculated the percent increases for suicidal ideation and SUD presentation by dividing the differential change in the post-period (i.e., the treatment effect estimate) by the difference in means from the pre-period to the post-period in the comparison group (i.e., the counterfactual). 

There is also a differential increase in the adjusted proportion of presentations with SUD in the COVID-19 time series relative to the comparison time series at all time points after the exposure. This differential increase is statistically significant at three weeks following the exposure, or March 23-27, 2020 (32.8 percentage points; 95% CI: 4.0, 61.6). Based on the comparison series, we would have expected a decrease of 6.2 percentage points for substance use at week three. Relative to what we would have expected based on diagnoses from the previous year during week three, we saw a 629.0% increase in SUD presentations following the coronavirus outbreak.”

9 In the Discussion section, the authors might consider integrating their findings with other studies regarding changes in volume of psychiatric cases associated with the COVID-19 pandemic.

Thank you for this valuable addition. We agree that highlighting the literature exploring changes in volume of psychiatric cases associated with the COVID-19 pandemic is important.

We have extended our discussion of the relevant literature in the Discussion section: “While another recent Massachusetts-based study suggests that overall suicide rates did not increase during the pandemic, it appears that the experience of suicidal ideation may have increased at certain time points following the outbreak (13). This finding regarding increased pandemic-related suicidal ideation, as well as our finding regarding increased presentations with SUD, is corroborated by another recent large-scale cross-sectional study (32).”

10. The study limitations section (pp. 15-16) makes some good observations. However, the large confidence intervals receive only fleeting reference, which does not really seem to do justice to this issue— it may be helpful to address this concern a bit more fully, including the ramifications of the very large CI’s for interpreting the findings.

Thank you for highlighting this important point. We agree, that emphasizing the large confidence intervals is important. However, after further examination and thought, we believe that these confidence intervals do not suggest that the study is underpowered (with the possible exception of the analysis with psychosis), as the treatment effect estimates do not appear to be unstably estimated zeroes. Nor do the confidence intervals preclude us from ruling out large effects in the opposite direction (i.e., in most instances, we can rule out large effects in one direction, but the effect may be larger than the point estimate). 

Additionally, we noticed that the size of the confidence intervals grows over the course of the post-period, which is likely an effect of using a general CITS. Because general CITS uses data to estimate a growing difference between the two groups, this design introduces greater uncertainty than designs that assume no growing difference (e.g., a difference-in-differences), and this uncertainty grows as estimates get further from the intervention and the linearity assumption does more work. 

We have attempted to address both points more fully in our Limitations sub-section in the Discussion section: “Fifth, while the confidence intervals for some of our results are large, they generally do not suggest that we are underpowered to detect an effect (i.e., an unstably estimated zero) with the exception of presentations with psychosis. In most cases, we can rule out large estimates in the opposite direction of the treatment effect, suggesting that the estimated treatment effects may be under-estimates of the increases or decreases seen in our study. Additionally, with a general CITS, estimates further from the intervention typically have larger confidence intervals, as uncertainty increases over the time period when estimating a between-group difference from the data.”

11. Table 2 refers to “mean (%)” in the first column, but evidently only percentages are reported— it would be useful to include the frequencies as well, and to specify how the percentages were calculated (percentage of total number of psychiatric cases during that interval, percentage of total [psychiatric and non-psychiatric] ED visits during that interval, etc.?).

Thank you for this suggestion – we have amended the table to indicate “n (%)” for the values reported. We also now specify how the percentages were calculated, with the n value in the column and the total n in the column head to indicate numerator and denominator respectively.

12. Table 1 reports percentages. It would be conventional to report the frequencies as well (though I appreciate that journals vary in their preferences and the editor may have a different view).

Thank you for this recommendation – we agree that inclusion of the n, in addition to the percentages, will be helpful. We now include the n values in Table 1.

I was unable to find Appendix in the material I reviewed.

Thank you – we now include three Appendix tables in the Supplementary Materials.

MINOR comments:

1. The use of the term “intervention” throughout the paper may be confusing to readers, since there is no intervention in the traditional clinical sense. Perhaps the authors might consider “exposure” instead.

We agree that the term “intervention” may be confusing to readers; we have changed this term to “exposure” throughout the manuscript. 

2. On p. 7 (lines 146-147), the text states that information was extracted “automatically” for some variables and “manually” for others. It is not entirely clear what this means (e.g., the medical records could be queried for summary information regarding some variables, but information had to be manually extracted by searching each record individually for other variables?).

Thank you for this helpful suggestion – we have amended the language in the Outcome measures sub-section in the Results section to clarify our data collection process: “In other words, the medical records could be queried for summary information for some variables, but information had to be manually extracted by searching each record individually for other variables.”

3. On p. 11 (lines 196-197), the text notes that Figure 3 summarizes the “adjusted models for each of the four psychiatric presentation groups.” It may be useful to specify the time period for these data (e.g., evidently the entire study period from Dec. to May).

We agree that specifying the study period will be useful. We have amended the paragraph following Table 2 in the Results section to make this clarification: “Our adjusted models for each of the four psychiatric presentation groups (suicidal ideation, affective disorders, psychotic disorders, and SUDs) for the two study periods, which run from December to May in 2018-2019 and 2019-2020, respectively, are summarized in Fig 3, with tabular data presented in the Appendix Table 2.” Thank you for this helpful suggestion.

4. On p. 11, in the last paragraph regarding changes in SI and SUDS, it would be helpful to reiterate that these analyses were adjusted (this was mentioned in the previous paragraph but not in this one).

Thank you for this thoughtful comment – we have amended the last two paragraphs of the Results section to clarify that we are discussing “adjusted” proportions.

5. On p. 3, line 57, it would be useful to note that the 30% in suicide rates associated with the 2003 SARS epidemic was reported in a study conducted in Hong Kong, if that is accurate (the previous clause refers to research in the US). Similarly, on p. 12 (lines 236), the last line on the page summarizes findings from the Faust et al. study-- it may be useful to specify that this investigation focused on rates in Massachusetts.

Thank you for this useful feedback. We now clarify that the 30% increase in suicide rates associated with the 2003 SARS epidemic was reported in a study conducted in Hong Kong. We also now specify that the Faust et al. study was conducted in Massachusetts.

6. PLOS authors have the option to publish the peer review history of their article (what does this mean?). If published, this will include your full peer review and any attached files.

Do you want your identity to be public for this peer review? For information about this choice, including consent withdrawal, please see our Privacy Policy.

Reviewer #1: No

Thank you for your thoughtful review of this manuscript.

---

## [Decision Letter · Decision Letter 1]

24 May 2021

PONE-D-21-02006R1

Evaluating the association between COVID-19 and psychiatric presentations, suicidal ideation in an emergency department

PLOS ONE

Dear Dr. McDowell,

Thank you for submitting your manuscript to PLOS ONE. After careful consideration, we feel that it has merit but does not fully meet PLOS ONE’s publication criteria as it currently stands. Therefore, we invite you to submit a revised version of the manuscript that addresses the points raised during the review process.

If applicable, we recommend that you deposit your laboratory protocols in protocols.io to enhance the reproducibility of your results. Protocols.io assigns your protocol its own identifier (DOI) so that it can be cited independently in the future. For instructions see: http://journals.plos.org/plosone/s/submission-guidelines#loc-laboratory-protocols. Additionally, PLOS ONE offers an option for publishing peer-reviewed Lab Protocol articles, which describe protocols hosted on protocols.io. Read more information on sharing protocols at https://plos.org/protocols?utm_medium=editorial-emailutm_source=authorlettersutm_campaign=protocols.

We look forward to receiving your revised manuscript.

Kind regards,

Corstiaan den Uil

Academic Editor

PLOS ONE

Journal Requirements:

Reviewers' comments:

Reviewer's Responses to Questions

**Comments to the Author**

1. If the authors have adequately addressed your comments raised in a previous round of review and you feel that this manuscript is now acceptable for publication, you may indicate that here to bypass the “Comments to the Author” section, enter your conflict of interest statement in the “Confidential to Editor” section, and submit your "Accept" recommendation.

Reviewer #1: (No Response)

2. Is the manuscript technically sound, and do the data support the conclusions?

Reviewer #1: Yes

3. Has the statistical analysis been performed appropriately and rigorously? 

Reviewer #1: Yes

4. Have the authors made all data underlying the findings in their manuscript fully available?

Reviewer #1: Yes

5. Is the manuscript presented in an intelligible fashion and written in standard English?

Reviewer #1: Yes

6. Review Comments to the Author

Reviewer #1: I think the authors have done a commendable job with the revision— the clarifications and additional information were very helpful. This paper will be a nice contribution to the literature in this area.

I have only a few remaining minor comments for the investigators’ consideration. The page numbers refer to the tracked version rather than the clean version:

1. As previously noted, the study aims section (p. 4, last paragraph of the Introduction) is a bit vague and it seems that outcomes are only clarified for the reader 5 pages later (p. 10). It might be useful to change “psychiatric complaints” to “certain psychiatric complaints, and then to note in parentheses “(a number of selected substance use, mood, and psychotic disorders or suicidal ideation)” or similar phrasing-- in other words, provide a clearer sense of the study endpoints. Just a thought.

2. The statistical analysis section (top of p. 7) indicates that “a general rather than a linear CITS” was used to reduce assumptions, but the subsequent newly added sentence states that “A general CITS assumes that any differential growth… is linear.” For readers unschooled in CITS, like myself, that might seem at first blush like a logical contradiction, and perhaps the authors might take another look at those 2 sentences for clarity. Additionally, the new text on the bottom of the page states that “patients were allowed to appear in more than one of these periods,” and it might be helpful to briefly assure readers that this does not introduce problematic dependency in the data.

3. In the Results section regarding findings for the post-exposure periods, I think it would be important to indicate more explicitly that separate analyses were run for each of 10 weeks, if I understand correctly. That is, the text describing the significant results for SI at six weeks (p. 15, line 282), and the passage describing significant results for SUD at three weeks (p. 16, line 302) should note more clearly that each of these findings represent one of 10 analyses for each outcome, if that is accurate. (This approach is mentioned in the Data Analysis section of the Methods but should be specified in the Results.) .

4. On p. 15 in the Results section, there is ambiguity regarding the type of analysis described in the 3rd sentence on the page (lines 281-285), since the 1st sentence refers to adjusted model and the 2nd sentence refers to the unadjusted analyses. It may be helpful to clarify for readers that the 3rd sentence is once again referring to the adjusted analyses. In addition, should the text mention that there was an unexpected marginally significant decrease in SI at week 9? This finding does not seem to be mentioned in the Results section, and instead emerges for the first time in the Discussion (p. 18, lines 235-360). Was there a significant decrease in SI for those presenting with psychotic disorders, and was there a significant increase in SI for those from “other” racial/ethnic backgrounds relative to whites (as seems to be suggested in Appendix 2)?

5. The new text in the Results section includes a helpful summary of changes in SI based on unadjusted analyses (p. 16). Should the text also mention that there was a significant increase in SI at week 6 in the unadjusted model (as suggested in Appendix 3)?

6. In Appendix Table 2 and 3, it is not clear whether the model as a whole was significant for each outcome (i.e., the sample size and R square is presented for each outcome in Table 2, which is great, but not the significance level for the overall model for each outcome). More generally, I had a bit of difficulty interpreting this table, no doubt due to my limited familiarity with CITS, but the authors might wish to explain this Appendix more clearly (e.g., how does comparison level differ from differential level?).

7. The Discussion section indicates that there was a differential decrease in SI at nine weeks after the exposure (p. 18, lines 357-360). It might be useful to clarify that the text is referring to the adjusted analyses.

Other MINOR comments:

1. It might be helpful to specify the CITS abbreviation the first time the term is used in the paper (e.g., insert “CITS” in parentheses after “Comparative interrupted time series” on p. 6 line 100).

2. The change in terminology from “intervention” to “exposure” is appreciated. To avoid confusion, the authors might wish to make the same change in the remainder of the manuscript (i.e., p. 6, lines 110-112; p. 8, line 150; p. 11, lines 207 and 225; p. 15, line 293; p. 22, lines 439-441; Appendix Table 3) though they

3. On p. 6, line 107, it may be helpful to change “psychiatric presentation” to “selected psychiatric presentations” or “certain psychiatric presentations” or comparable language. On p. 11, the authors might wish to change the phrasing in the first and last paragraphs to the past tense.

4. On p. 12, for clarity, it would be useful to reiterate the time intervals for readers, e.g., by inserting in parentheses “(Dec. 2019- Feb. 2020)” after “COVID-19 pre-period” and by inserting “(Dec. 2018- Feb. 2019)” after “comparator pre-period.”

5. On p. 18, line 355, to facilitate interpretation it may be helpful to note the location of the study cited in this sentence (e.g., perhaps insert “in the US” after “cross-sectional study”).

6. On p. 20, it may be helpful to clarify the direction of the effects (perhaps by changing “White” to “fewer White,” and changing “Asian” to “more Asian” on line 397, and by inserting “in the 2019-2020 cohort” after “with affective disorders” on line 398).

7. Appendix Table 3, it may be useful to specify the demographic/temporal covariates that were included in all these analyses, in a note at the bottom of the table, to avoid a misimpression. In the title, after “Differential change in suicidal ideation presentations,” the authors might consider inserting “from ____ to ____” (i.e., specify the comparison conditions).

Good luck with your work!

7. PLOS authors have the option to publish the peer review history of their article (what does this mean?). If published, this will include your full peer review and any attached files.

Reviewer #1: No

---

## [Author Response · Author response to Decision Letter 1]

8 Jun 2021

Reviewer #1: I think the authors have done a commendable job with the revision— the clarifications and additional information were very helpful. This paper will be a nice contribution to the literature in this area.

Many thanks for your generous comments on our manuscript! We believe they have significantly improved the quality of the work we submit now.

I have only a few remaining minor comments for the investigators’ consideration. The page numbers refer to the tracked version rather than the clean version:

1. As previously noted, the study aims section (p. 4, last paragraph of the Introduction) is a bit vague and it seems that outcomes are only clarified for the reader 5 pages later (p. 10). It might be useful to change “psychiatric complaints” to “certain psychiatric complaints, and then to note in parentheses “(a number of selected substance use, mood, and psychotic disorders or suicidal ideation)” or similar phrasing-- in other words, provide a clearer sense of the study endpoints. Just a thought.

We agree that our aims in the last paragraph of the introduction could use clarification. We appreciate your suggestions and have included the word “certain” to qualify “psychiatric disorders.” We also now indicate in parentheses the outcomes of interest: “We used the period following the initial outbreak of COVID-19 in Massachusetts to assess the association between the pandemic and patients presenting with certain psychiatric complaints (a number of selected mood, psychotic, and substance use disorders, as well as suicidal ideation) to a tertiary care hospital Emergency Department (ED).”

2. The statistical analysis section (top of p. 7) indicates that “a general rather than a linear CITS” was used to reduce assumptions, but the subsequent newly added sentence states that “A general CITS assumes that any differential growth… is linear.” For readers unschooled in CITS, like myself, that might seem at first blush like a logical contradiction, and perhaps the authors might take another look at those 2 sentences for clarity. Additionally, the new text on the bottom of the page states that “patients were allowed to appear in more than one of these periods,” and it might be helpful to briefly assure readers that this does not introduce problematic dependency in the data.

Thank you for highlighting these points, which we agree would all benefit from clarification. 

A linear CITS assumes that the evolution of the outcome (i.e., the probability of a patient showing up to the ED with a psychiatric presentation) is linear over the course of the study, which makes a parametric assumption about the post-period outcomes (i.e., that it follows the form of a line). However, a general CITS does not make an assumption about the post-period outcomes follow any shape (liner, quadratic, etc.). In a general CITS, the outcome is not assumed to be of a specific functional form. Instead, general CITS makes the assumption that the difference between the two groups is evolving linearly. 

To clarify this point to our readers, we have included the following changes in the manuscript text in the second full paragraph of the Methods section, Study design sub-section: “Finally, we use a general CITS, rather than a linear CITS, to minimize the parametric assumptions made regarding the outcome’s evolution. A linear CITS assumes that the outcome is evolving linearly in the pre- and post-period (24). On the other hand, a general CITS assumes that any difference between the two groups is evolving linearly but does not make any assumption about the outcome’s parametric form (i.e., does not assume a linear, quadratic, or other form). We conducted an event study to visually assess the plausibility of these assumptions and found differential trends (i.e., non-parallel) in the pre-period suggesting that a standard difference in difference approach would not be appropriate.”

Regarding the issue of multiple presentations, we treat each of the four periods (2018-2019 pre, 2018-2019 post, 2019-2020 pre, 2019-2020 post) as a cohort in this cross-sectional study. We do make adjustments for patients presenting multiple times in one of the study periods, which we now specify in the antepenultimate paragraph in the Methods section, Study design sub-section: “We adjusted the models for the total number of emergency department visits per patient per study period.” 

3. In the Results section regarding findings for the post-exposure periods, I think it would be important to indicate more explicitly that separate analyses were run for each of 10 weeks, if I understand correctly. That is, the text describing the significant results for SI at six weeks (p. 15, line 282), and the passage describing significant results for SUD at three weeks (p. 16, line 302) should note more clearly that each of these findings represent one of 10 analyses for each outcome, if that is accurate. (This approach is mentioned in the Data Analysis section of the Methods but should be specified in the Results.)

Thank you for this helpful suggestion. We agree that this point certainly needs clarification. 

We did not run a separate analysis for each of the 10 weeks in the post-period. Rather, our regression specification includes an indicator variable for each post-period week. We include the indicator variable and an interaction between the treatment variable and the weekly indicator variable. Thus, each of these interactions represent the differential change for this week between the COVID-19 series and comparator series from the average pre-period level. We include our single regression formula at the beginning of the manuscript’s Supplemental Material to provide further explanation of the regression analysis conducted. In addition, we removed the “s” from “general CITS regression(s)” at the beginning of our Supplemental materials section to clarify that the analyses presented derive from a single regression specification. 

We have added a sentence in the last paragraph of the Methods section, Data analysis sub-section to further clarify this point, as well: “In other words, our regression specification included an indicator variable for each post-period week. We and interacted the indicator variable with the treatment variable, and each of these interactions represent the differential change for the week between the COVID-19 series and comparator series as compared to the average pre-period levels.” 

4. On p. 15 in the Results section, there is ambiguity regarding the type of analysis described in the 3rd sentence on the page (lines 281-285), since the 1st sentence refers to adjusted model and the 2nd sentence refers to the unadjusted analyses. It may be helpful to clarify for readers that the 3rd sentence is once again referring to the adjusted analyses. In addition, should the text mention that there was an unexpected marginally significant decrease in SI at week 9? This finding does not seem to be mentioned in the Results section, and instead emerges for the first time in the Discussion (p. 18, lines 235-360). Was there a significant decrease in SI for those presenting with psychotic disorders, and was there a significant increase in SI for those from “other” racial/ethnic backgrounds relative to whites (as seems to be suggested in Appendix 2)?

We agree that the first sentences in the Results section paragraph following Fig 3 were confusing to follow. We have improved clarity by reordering the sentences such that we mention the unadjusted models first, and then discuss the adjusted models following. 

Thank you for the suggestion that we name the non-significant decrease in SI at week 9 in the Results section. We now do this at the end of the Results section paragraph following Fig 3: “Additionally, at nine weeks after the exposure (May 4-9, 2020), there was a differential decrease in the proportion of psychiatric presentations to the ED with suicidal ideation (34.9 percentage points; 95% CI; -69.5, -0.3), though this change was not statistically significant (p = 0.06).” 

Your questions regarding Appendix Table 2 are helpful. In the “Any SUD,” “Any affective disorder,” “Any psychotic disorder” and “Other” race/ethnic background rows, you are correct that the confidence intervals do not cross 0 for the suicidal ideation column. This indicates that, over the course of the study, after accounting for all other covariates, including the exposure (COVID-19 outbreak), there is a relationship between suicidal ideation and other psychiatric conditions and/or demographic characteristics. In other words, the probability of presentation with suicidal ideation increases with the presence of co-occurring SUD, increases with co-occurring affective disorders, decreases with co-occurring psychotic disorders, and increases in people with “other” race/ethnic backgrounds. These changes do not have do to with COVID. 

We are happy to provide additional text clarifications on Appendix Table 2 at the Editor and Reviewer’s discretion. 

5. The new text in the Results section includes a helpful summary of changes in SI based on unadjusted analyses (p. 16). Should the text also mention that there was a significant increase in SI at week 6 in the unadjusted model (as suggested in Appendix 3)?

Thank you for highlighting this point, which could certainly use clarification. We now include more explanation regarding Appendix Table 3 in the paragraph following Fig 3 in the Results section: “We present data in Appendix Table 3 for suicidal ideation, sequentially removing and re-adding other certain psychiatric presentations to the model. Our results change modestly as a result of this different regression specification. Specifically, there is a significant increase in presentations with suicidal ideation at weeks three and six. However, once adjustment for SUD is included in the model at week three, the differential change is no longer significant because this increase was mediated by the co-occurring SUD diagnoses. However, even after the adjustments for co-occurring psychiatric conditions are added at week six, the differential change persists.”

6. In Appendix Table 2 and 3, it is not clear whether the model as a whole was significant for each outcome (i.e., the sample size and R square is presented for each outcome in Table 2, which is great, but not the significance level for the overall model for each outcome). More generally, I had a bit of difficulty interpreting this table, no doubt due to my limited familiarity with CITS, but the authors might wish to explain this Appendix more clearly (e.g., how does comparison level differ from differential level?).

We agree that Appendix Tables 2 and 3 may be challenging to follow. 

First, we now include the global F-test statistic, p-value, and degrees of freedom to Appendix Table 2 and 3. In Appendix Table 2, the F-test statistics demonstrate that for suicidal ideation, affective disorders, and SUD, the models are statistically significant. For psychotic disorders, the overall model is not statistically significant, suggesting that this model does not explain the variation in the outcome better than the intercept-only model, suggesting that changes in the presence of psychotic disorder may not be attributable to COVID-19 and may be more complex than the current regression models can explain. 

Second, we have re-arranged Appendix Table 2 to first display the quantities of interest (i.e., the treatment effects).

Third, we have added text in the notes of both Tables to help readers understand what these estimates represent: “This is the difference between the change in proportion of ED visits with a psychiatric presentation from the pre-period to post-period for the COVID-19 series and the change in the proportion of ED visits with a psychiatric presentation from the pre-period to the post-period in the comparison series (quantity of interest).”

7. The Discussion section indicates that there was a differential decrease in SI at nine weeks after the exposure (p. 18, lines 357-360). It might be useful to clarify that the text is referring to the adjusted analyses.

We agree that this clarification would prove helpful. We now include the qualifier “in the adjusted analyses,” in this sentence. 

Other MINOR comments:

1. It might be helpful to specify the CITS abbreviation the first time the term is used in the paper (e.g., insert “CITS” in parentheses after “Comparative interrupted time series” on p. 6 line 100).

Thank you for this catch! We now include the abbreviation after the first time the term is used in the paper. 

2. The change in terminology from “intervention” to “exposure” is appreciated. To avoid confusion, the authors might wish to make the same change in the remainder of the manuscript (i.e., p. 6, lines 110-112; p. 8, line 150; p. 11, lines 207 and 225; p. 15, line 293; p. 22, lines 439-441; Appendix Table 3) though they

Thank you for this helpful suggestion – we have changed the terms from “intervention” to “exposure” throughout the manuscript.

3. On p. 6, line 107, it may be helpful to change “psychiatric presentation” to “selected psychiatric presentations” or “certain psychiatric presentations” or comparable language. On p. 11, the authors might wish to change the phrasing in the first and last paragraphs to the past tense.

Thank you, we agree these clarifications would be useful for readers. We have amended to include “certain.” We have also amended the two indicated paragraphs to read in the past tense.

4. On p. 12, for clarity, it would be useful to reiterate the time intervals for readers, e.g., by inserting in parentheses “(Dec. 2019- Feb. 2020)” after “COVID-19 pre-period” and by inserting “(Dec. 2018- Feb. 2019)” after “comparator pre-period.”

Thank you for this helpful suggestion. We now include the following clarifications: In exploring pre-period differences, we found that the COVID-19 pre-period (December 23, 2019 – February 24, 2020) cohort had a higher proportion of psychiatric ED consultations involving people from a minority racial group than did the comparator pre-period (December 24, 2018 – February 25, 2019) (Table 1).”

5. On p. 18, line 355, to facilitate interpretation it may be helpful to note the location of the study cited in this sentence (e.g., perhaps insert “in the US” after “cross-sectional study”).

We agree that this would be a helpful clarification. The sentence now reads: “This finding regarding increased pandemic-related suicidal ideation, as well as our finding regarding increased presentations with SUD, is corroborated by another recent large-scale cross-sectional study in the United States (32).”

6. On p. 20, it may be helpful to clarify the direction of the effects (perhaps by changing “White” to “fewer White,” and changing “Asian” to “more Asian” on line 397, and by inserting “in the 2019-2020 cohort” after “with affective disorders” on line 398).

We certainly agree that this clarification would be helpful to readers. We have made the suggested edits and the sentence now reads: “When we further explore this difference by separating the groups by psychiatric presentations, statistically significant differences are isolated to fewer White patients presenting with psychotic disorders and more Asian patients presenting with affective disorders in the 2019-2020 cohort.”

7. Appendix Table 3, it may be useful to specify the demographic/temporal covariates that were included in all these analyses, in a note at the bottom of the table, to avoid a misimpression. In the title, after “Differential change in suicidal ideation presentations,” the authors might consider inserting “from ____ to ____” (i.e., specify the comparison conditions).

Thank you for this helpful suggestion. We have amended the title of the table to better represent what we are showing in the table and have added notes to the table to further explain our covariate adjustment and the meaning of the coefficients in the table. 

Good luck with your work!

Many thanks for your patience and generosity in reviewing and offering suggestions to our paper.

---

## [Editor Report · Decision Letter 2]

14 Jun 2021

Evaluating the association between COVID-19 and psychiatric presentations, suicidal ideation in an emergency department

PONE-D-21-02006R2

Dear Dr. McDowell,

We’re pleased to inform you that your manuscript has been judged scientifically suitable for publication and will be formally accepted for publication once it meets all outstanding technical requirements.

Kind regards,

Corstiaan den Uil

Academic Editor

PLOS ONE
---

## [Editor Report · Acceptance letter]

23 Jun 2021

PONE-D-21-02006R2 

Evaluating the association between COVID-19 and psychiatric presentations, suicidal ideation in an emergency department 

Dear Dr. McDowell:

I'm pleased to inform you that your manuscript has been deemed suitable for publication in PLOS ONE. Congratulations! Your manuscript is now with our production department. 

Kind regards, 

on behalf of

Dr. Corstiaan den Uil 

Academic Editor

PLOS ONE